# ECHOQA: TUNING INTO THE HEART OF ECHOCARDIOGRAM REPORTS

## ABSTRACT

We introduce a novel and extensive question-answering dataset using echocardiogram reports sourced from Medical Information Mart for Intensive Care (MIMIC) data. This dataset is specifically tailored to enhance question answering (QA) systems in the field of cardiology. It comprises 765,605 QA pairs addressing a wide array of cardiac abnormalities and their severity. To validate the utility of this benchmark dataset, we employ various large language models (LLMs), encompassing both open-source general models and biomedical-specific models, along with state-of-the-art closed-source models for zero-shot evaluation. Our results reveal that certain models achieve superior performance across all evaluated metrics. Further, we conduct a fine-grained fairness audit of the best performing LLM across demographic groups and marginalized populations. Our objective is to propel the field forward by establishing a benchmark framework for developing LLM AI agents that support clinicians in their daily workflow within the cardiology space. The availability of this dataset aims to support the advancement of natural language models for use in diagnostic decision support systems, aiming to increase efficiency in cardiology care. All code is available at https://anonymous.4open.science/r/echoqa-02E3/README.md and the data will be made available on HIPAA-compliant data repository PhysioNet.

## 1 INTRODUCTION

Echocardiography is the most prevalent noninvasive technique for assessing heart function and detecting heart diseases. It plays a critical role in clinical cardiology, consistently guiding decision-making processes (Nagueh et al., 2016). Echocardiography is essential for diagnosing diseases, stratifying risks, and evaluating treatment efficacy. The diagnostic reports generated from these tests provide rich clinical data, vital for diagnosing and managing various cardiac conditions (Lang et al., 2015). However, the large volume and complexity of these reports pose significant challenges for clinicians, potentially causing delays in decision-making and increasing their cognitive load. Furthermore, the growing demand for diagnostic echocardiograms exacerbates these challenges, making it even more difficult to manage and interpret the increasing volume of data.

The advent of large language models (LLMs) holds the potential to transform the field of cardiology. LLMs have been utilized across various natural language processing tasks, such as question-answering (QA), text summarization, and language translation, often in zero-shot scenarios without the need to update model parameters (Brown et al., 2020). Moreover, converting natural language understanding and generation tasks into instructional inputs enhances LLMs' ability to follow domain-specific instructions and improve downstream task performance (Wei et al., 2021; Ouyang et al., 2022). Open-source models like LLaMA (Touvron et al., 2023) and Mistral (Jiang et al., 2023) have shown great potential. Leveraging these models with high-quality instruction-following samples from echocardiogram diagnostic reports is important to streamlining healthcare workflows. Such systems would efficiently handle the complexity of data, reduce clinicians' cognitive load, and facilitate faster decision-making.

A significant challenge has been the development of LLMs trained and evaluated on real echocardiogram reports with ground-truth answers, rather than relying on synthetic data or those from medical licensing exams (Zhang et al., 2018; Kweon et al., 2023). This limitation has hindered progress of AI in the cardiology space. However, with the recent advancements in instruction-following capa-

bilities of LLMs, there is an opportunity to bridge the gap between raw data and actionable medical insights and assist with clinical reasoning and knowledge recall. A high-quality question-answering dataset using echocardiogram reports could serve as a benchmark and allow the training of specialized cardiology models, reducing healthcare providers' cognitive load while streamlining workflows to increase efficiency.

Addressing algorithmic discrimination is crucial in healthcare. Utilizing protective attributes, such as race, gender and age can improve the effectiveness of fairness audits in healthcare algorithms (Obermeyer et al., 2019; Rajkomar et al., 2018). Beyond these common attributes, analyzing social factors such as disability and lifestyle behaviors could assist healthcare providers identify and mitigate disparities in algorithmic outcomes. These living conditions offer valuable insights for addressing gaps in medical care. Such non-clinical factors, commonly known as social determinants of health (SDOH), are vital for conducting thorough audits of algorithmic biases (Wilkinson, 2003). Ensuring that algorithms account for the broader context of patients' lives promotes equitable healthcare, especially for marginalized groups (Moukheiber et al.). Incorporating these data into fairness audits also aids in complying with regulations like Section 1557 of the Affordable Care Act, which mandates that healthcare providers and payers ensure their algorithms do not discriminate (Cary Jr et al., 2023).

Based on the challenges aforementioned, our work makes the following three contributions:

- *Development of EchoQA:* We present EchoQA, the largest open-access, real patient question-answering dataset for echocardiography, meticulously developed by expert clinicians. Our aim is to propel the medical field by creating a foundation for training LLM-based AI agents that will assist cardiologists in their daily workflows. EchoQA offers a robust dataset that offers researchers and practitioners to test and compare different machine learning approaches for disease diagnosis and management. This resource, along with the accompanying code, will be freely available on PhysioNet, an open-source healthcare data repository.

- *Instruction Fine-Tuning and Zero-Shot Evaluations:* Leveraging the EchoQA dataset, we showcase its potential by fine-tuning various LLMs, including both general and medical-domain models. Additionally, we conduct zero-shot evaluations on high-profile commercial LLMs such as OpenAI GPT-4o, Amazon Titan (Amazon Web Services, 2024), Claude (Anthropic, 2024), and Cohere (Cohere, 2024). Furthermore, we will release the best-performing echocardiogram model, Echo-Mistral, making it accessible to the wider research community.

- *Fairness Audits on Social Factors:* To investigate algorithmic bias, we use SDOH to enable more fine-grained audits of algorithmic fairness. These evaluations provide critical insights into potential disparities often overlooked in LLM studies, ensuring a more equitable and inclusive approach to medical artificial intelligence.

## 2 RELATED WORK

**Medical question answering datasets.** Medical question-answering benchmark datasets have been developed to address different aspects of medical information retrieval and understanding. For instance, MedQA is designed to focus on Chinese medical licensing examination questions, mimicking real-world medical exams and educational tools (Zhang et al., 2018). PubMedQA includes biomedical research questions derived from PubMed abstracts, facilitating research question understanding (Jin et al., 2019). The emrQA dataset consists of over 400,000 factual questions with answers provided in electronic medical records, enhancing the understanding of clinical data (Pampari et al., 2018). LiveQA includes questions users ask online in real-time, offering insights into immediate medical information needs (Liu et al., 2020). MedicationQA focuses on questions related to nearly 700 medications and their uses, aiding in pharmaceutical information retrieval (Abacha et al., 2019). MMLU Clinical Topics is part of the Massive Multitask Language Understanding (MMLU) benchmark and includes a section on clinical topics, supporting broader medical knowledge evaluation (Hendrycks et al., 2020). HealthSearchQA is a newly introduced dataset consisting of 3,173 commonly searched consumer medical questions, capturing the health-related inquiries typically asked in search engines and reflecting common health concerns of the general public (Singhal et al., 2023). It is not known if performance on these benchmark datasets will translate when a model

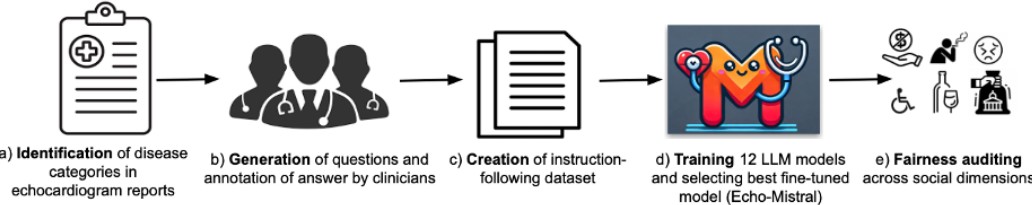

Figure 1: EchoQA Workflow. (a) We identify cardiac abnormalities and severities in echocardiogram reports. (b) We prepare question and answers verified with clinical experts. (c) We create an echo-cardiogram question-answering dataset. (d) We use 10,000 randomly sampled question and answering pairs to train and evaluate the data using twelve LLM methods. (e) We perform fairness audits across social factors to assess bias.

is deployed in the complex clinical environments. However, none of the current datasets leverage question-answer pairs curated from a large cohort of patient data in the cardiology domain, hindering advancement in the application of these models to real-world clinical settings.

**Fairness audits.** While progress has been made in addressing algorithmic fairness in healthcare, most studies have focused primarily on biases related to protected attributes such as age, gender, and race (Obermeyer et al., 2019; Chen et al., 2019; Seyyed-Kalantari et al., 2020; Zhang et al., 2022). Recent research emphasizes the need to examine biases from more multidimensional perspectives, particularly by analyzing the social processes that contribute to these biases. Evaluating fairness through the lens of intersectional social identities provides a deeper understanding of the socially constructed nature of attributes like race and gender. Incorporating social factors offers valuable insights into the processes driving disparities. Furthermore, conducting bias audits centered on these factors is more practical, as they are not just social constructs but modifiable aspects (Braveman & Gottlieb, 2014; Chen et al., 2020). To the best of our knowledge, we are also the first to leverage social determinants of health to conduct fine-grained audits of algorithmic fairness on biomedical and closed source LLMs. With that, we hope to ensure that the models are equitable and account for the broader context of individuals' lives.

## 3 EXPERIMENTAL SETUP

### 3.1 DATASET AND FRAMEWORK

Figure 1 illustrates the pipeline of our study. We curate a comprehensive question-answering dataset sourced from the Medical Information Mart for Intensive Care (MIMIC-IV) database, which is a de-identified clinical dataset comprising over 80,000 echocardiogram reports collected at Beth Israel Deaconess Medical Center between 2012-2019 Johnson et al. (2023).

To develop the question-answering system we consulted with clinical experts. Figure 2 depicts the structure of the echocardiogram report and the respective question and answer pair constructed for that echocardiogram report sample.

Clinicians formulated questions encompassing various aspects such as valvular evaluation, chamber size, and function evaluation. These covered categories including aortic valve regurgitation, aortic valve stenosis, left atrial cavity, left ventricular cavity, left ventricular diastolic function, left ventricular systolic function, left ventricular wall thickness, mitral valve regurgitation, mitral valve stenosis, right atrial pressure, right ventricular cavity, right ventricular systolic function, right ventricular wall thickness, tricuspid valve pulmonary hypertension, tricuspid valve regurgitation, and tricuspid valve stenosis.

To determine the severity of the abnormalities, questions were formulated to evaluate whether the study is adequate, if an abnormality is present, and whether the abnormality is quantifiable. The abnormalities are divided into severity levels such as mild, moderate, and severe. For right atrial dilation, right ventricular pressure overload, and right ventricular volume overload, the study included questions on whether it is appropriate to assess for these abnormalities and whether these abnormalities are present. For left ventricular cavity size and right ventricular cavity size, questions

included whether dilated cavities where present or if the cavity size was unusually small. Similarly, in cases of left ventricular systolic function, hyperdynamic systolic function was also recorded as an abnormality. After compiling and retrieving these categories, we programmatically curated a text analysis pipeline to enable the categorization and extraction of the question-answer pairs from the reports. This resulted in more than 700,000 question-answer pairs, with the categories depicted in Table 1. Evaluating the cardiac categories and parameters obtained from the diagnostic reports is crucial in clinical practice as they provide essential information about the structure and function of the heart, which is important for diagnosing and managing various cardiovascular diseases. Accurate assessment of these abnormalities helps guide therapeutic decisions and predict patient outcomes. The dataset will be hosted on a PhysioNet, an NIH-funded health data repository (Goldberger et al., 2000) and access will require user credentialing, including completing CITI ethics training and agreeing to the terms of the data use agreement.

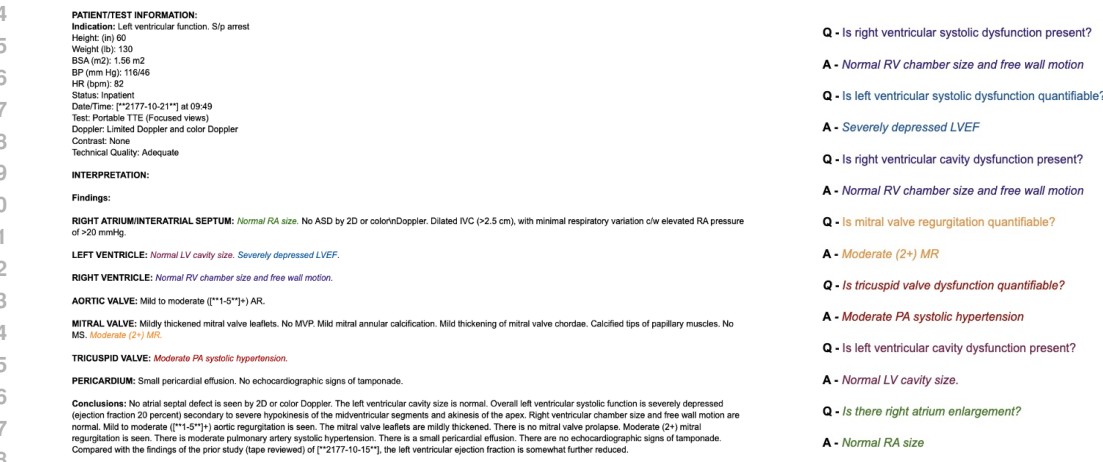

Figure 2: An example of a sample of an echocardiogram report and the question and answer pairs annotated by clinical experts.

## 4 DATA AND MODEL

We randomly sample 10,000 question-answer pairs from the curated dataset. The data is split into training, validation, and test datasets with a split of 70%, 20%, and 10%, respectively, ensuring no patient overlap to avoid data leakage. We used standardized prompts, where we prompted the models with a base instruction, ("Below is an echocardiogram report followed by a question"), followed by the echocardiogram report passage, and the respective question, ("Write an answer by extracting it from the report").

### 4.1 SUPERVISED FINE-TUNING (SFT) AND ZERO-SHOT VALIDATION

We employ supervised fine-tuning (SFT) to fine-tune our models using a diverse selection from recent open-source and biomedical domain-specific large language models (LLMs). From the open-source category, we utilize Llama-3-8b (Touvron et al., 2023), Mistral-7B (Jiang et al., 2023), Phi-3-mini (Abdin et al., 2024), and Zeyphr-7b Tunstall et al. (2023). For the biomedical open-source models, we incorporated BioMistral-7B Labrak et al. (2024), M42-health (Christophe et al., 2024), PMCLlama-13b (Wu et al., 2024), and Meditron-7B (Chen et al., 2023) aiming to leverage their deep understanding of biomedical terminology and context derived from medical abstracts and texts. Additionally, we use the propriety models, including Amazon-titan (Amazon Web Services, 2024), Claude (Anthropic, 2024), Cohere (Cohere, 2024), GPTo (Achiam et al., 2023) models for zero-shot evaluation. These models offer advanced capabilities in text generation and comprehension tasks without further fine-tuning. The closed-source models are run on Azure OpenAI and Amazon Bedrock to ensure HIPAA compliance and patient privacy.

Training is conducted for one epoch with a learning rate of 2e-5, employing a cosine learning rate schedule and a 1% warm-up ratio to stabilize the initial training phases. We use an Adam optimizer

| Cardiac Abnormalities | # of QA's |
|---|---|
| **Right Atrial abnormalities** | |
| Right Atrial Cavity | 45,262 |
| **Tricuspid Valve Abnormalities** | |
| Tricuspid Valve Regurgitation | 13,332 |
| Tricuspid Valve Stenosis | 19,509 |
| Pulmonary Hypertension | 21,136 |
| **Right Ventricular Abnormalities** | |
| Right Ventricular Systolic Function | 74,302 |
| Right Ventricular Cavity | 72,003 |
| Right Ventricular Volume Overload | 5,071 |
| Right Ventricular Pressure Overload | 5,495 |
| Right Ventricular Wall | 7,295 |
| **Left Atrial abnormalities** | |
| Left Atrium Cavity | 22,525 |
| **Mitral Valve Abnormalities** | |
| Mitral Valve Stenosis | 38,052 |
| Mitral Valve Regurgitation | 53,270 |
| **Left Ventricular abnormalities** | |
| Left Ventricular Systolic Function | 64,461 |
| Left Ventricular Cavity | 64,355 |
| Left Ventricular Wall | 64,276 |
| **Aortic Valve Abnormalities** | |
| Aortic Valve Stenosis | 61,422 |
| Aortic Valve Regurgitation | 59,626 |
| **Total** | **765,605** |

Table 1: Cardiac Abnormalities found in the MIMIC-IV echocardiogram reports.

with $\beta_1$ and $\beta_2$ parameters set to 0.9 and 0.95, respectively, and an epsilon value of 1e-5 to ensure numerical stability. The fine-tuning process is executed on NVIDIA A100 80GB GPUs, with each training session taking approximately 2-4 hours with model sharding. We utilize Low-Rank Adaptation (LoRA) based parameter- efficient fine-tuning as it enables the adaptation of models with minimal additional parameters, making it an efficient method for customizing the models to specific tasks. We also used BitsandBytes (BnB) quantization technique. BnB quantization assigns a fixed precision of 4 bits to the entire model, reducing the model size and computational load allowing a high-performance LLMs on hardware with limited capacity without sacrificing significant accuracy.

## 4.2 EVALUATION

We conduct a comprehensive analysis of our model's performance using a suite of well-established metrics. BLEU score is employed to measure the precision of n-grams between the generated and reference answers, providing insights into the accuracy of the text generation (Papineni et al., 2002). To assess the balance between precision and recall, we utilized the average F1 Score (Zhang et al., 2015). The ROUGE-1 and ROUGE-2 metrics are applied to evaluate the overlap of unigrams and bigrams, respectively, between the generated and reference answers, thereby gauging the lexical similarity at different granularities (Lin & Och, 2004). Additionally, the ROUGE-L metric is used to measure the longest common subsequence, indicating the extent to which the generated answer aligns with the reference in terms of sequence matching (Barbella & Tortora, 2022). Lastly, we utilize the average METEOR Score, which evaluates precision and recall while considering synonyms and stemming (Banerjee & Lavie, 2005).

We conduct fairness audits by examining social health attributes, as these factors provide insights into the conditions in which individuals live —critical influences on a person's health and well-being. To perform these audits, we utilize census tract-level social determinants of health data from the MIMIC dataset (Yang et al., 2023). Our analysis investigates fairness disparities across subgroups defined by societal attributes, such as whether a patient lives in areas with high unemployment rates, relies heavily on public assistance or food stamps, includes adults who are heavy drinkers or smokers, or reports experiencing mental distress or having a disability. We discretize the dimensions into high, upper middle, lower middle, low groups, based on the quantile of the distribution for each

dimension. For each LLM model, to assess bias across various dimensions, we use F1 equality difference in (Mansfield et al., 2022), which measures the average absolute difference between the f1 of individual social groups and the overall f1 across all groups within the corresponding social category. In particular, for a dimension $D$ and its associated set of demographic groups $\mathcal{G}^D = \{\mathcal{G}_1^D, \mathcal{G}_2^D, \dots\}$, F1 equality difference $= \frac{1}{|\mathcal{G}^D|} \sum_{\mathcal{G}_i^D \in \mathcal{G}^D} |\text{F1}(\mathcal{G}_i^D) - \text{F1}(\mathcal{G}^D)|$.

## 5 RESULTS & DISCUSSION

| Evaluation Metric | BLEU | ROUGE-1 | ROUGE-2 | ROUGE-L | F1 | METEOR |
|---|---|---|---|---|---|---|
| **Open-source (biomedical)** | | | | | | |
| BioMistral-7B zero-shot | 0.040196 | 0.248973 | 0.126855 | 0.228080 | 0.261562 | 0.334309 |
| BioMistral-7B | **0.673920** | **0.983662** | **0.979752** | **0.983533** | **0.983711** | **0.969618** |
| M42-health zero-shot | 0.198073 | 0.632796 | 0.531261 | 0.605784 | 0.636379 | 0.661092 |
| M42-health | 0.332752 | 0.755804 | 0.703983 | 0.748216 | 0.756412 | 0.748396 |
| Meditron7B zero-shot | 0.000147 | 0.078534 | 0.022602 | 0.071774 | 0.080893 | 0.058316 |
| Meditron7B | 0.358463 | 0.649220 | 0.571551 | 0.628676 | 0.654800 | 0.677095 |
| PMC-llama-13B zero-shot | 0.004293 | 0.097565 | 0.016630 | 0.092605 | 0.099424 | 0.073629 |
| PMC-llama-13B | 0.011070 | 0.140292 | 0.049413 | 0.128189 | 0.144872 | 0.137687 |
| **Open-source (general)** | | | | | | |
| Llama-8B-3.1 zero-shot | 0.077861 | 0.279651 | 0.221164 | 0.272437 | 0.307062 | 0.488860 |
| Llama-8B-3.1 | 0.598941 | 0.973055 | 0.968103 | 0.972872 | 0.973119 | 0.953933 |
| Mistral-7B zero-shot | 0.062683 | 0.296326 | 0.185250 | 0.272155 | 0.272155 | 0.312233 |
| Mistral-7B | **0.676927** | **0.984996** | **0.982060** | **0.984830** | **0.984830** | **0.985003** |
| Phi-mini zero-shot | 0.032677 | 0.230147 | 0.098855 | 0.206622 | 0.236978 | 0.264780 |
| Phi-mini | 0.594570 | 0.912221 | 0.890595 | 0.911864 | 0.912219 | 0.886837 |
| Zephyr-7B zero-shot | 0.056972 | 0.241928 | 0.164638 | 0.227090 | 0.227090 | 0.259500 |
| Zephyr-7B | 0.669497 | 0.980017 | 0.977374 | 0.980017 | 0.980017 | 0.980030 |
| **Closed-source (general)** | | | | | | |
| Amazon-titan | **0.234879** | **0.651269** | **0.542347** | **0.621926** | **0.654333** | **0.689966** |
| Claude | 0.091031 | 0.319220 | 0.259368 | 0.315331 | 0.341031 | 0.536380 |
| Cohere | 0.073882 | 0.394789 | 0.258828 | 0.364476 | 0.397607 | 0.396928 |
| GPT-4o | 0.139958 | 0.487212 | 0.396132 | 0.471600 | 0.496125 | 0.612211 |

Table 2: Performance metrics for open-source biomedical models, open-source general models, and closed-source general models averaged across 3 runs (higher scores are better). Fine-tuned open source models are compared to their baseline zero-shot models. Bolded numbers depict the best model.

| SDOH Attributes (*) | Disabled | Public Assistance | Unemployed | Heavy Drinkers | Poor Mental Health | Smokers |
|---|---|---|---|---|---|---|
| **Open-source (biomedical)** | | | | | | |
| BioMistral-7B | 0.005297 | **0.003684** | 0.006792 | **0.006104** | **0.007288** | 0.004340 |
| M42-health | 0.010288 | 0.008083 | 0.007291 | 0.013292 | 0.022794 | 0.025824 |
| Meditron-7B | 0.013354 | 0.016789 | 0.013501 | 0.022763 | 0.019980 | 0.008858 |
| PMC-llama-13B | **0.005274** | 0.010131 | **0.003061** | 0.015347 | 0.008790 | **0.003494** |
| **Open-source (general)** | | | | | | |
| Llama-8B-3.1 | 0.008496 | 0.008939 | 0.004903 | **0.002734** | 0.010982 | 0.017511 |
| Mistral-7B | **0.002104** | **0.001602** | **0.002772** | 0.003352 | **0.003430** | 0.008934 |
| Phi-mini | 0.008543 | 0.009321 | 0.011790 | 0.008246 | 0.015987 | 0.016267 |
| Zephyr-7B | 0.009294 | 0.007508 | 0.004649 | 0.005622 | 0.008858 | **0.006582** |
| **Closed-source (general)** | | | | | | |
| Amazon-titan | 0.012920 | **0.004267** | 0.011180 | 0.018461 | 0.014990 | 0.016885 |
| Claude | 0.009282 | 0.006137 | **0.008851** | **0.007393** | **0.003030** | **0.010465** |
| Cohere | 0.015901 | 0.011245 | 0.012095 | 0.019227 | 0.015437 | 0.019973 |
| GPT-4o | **0.008103** | 0.015158 | 0.012009 | 0.013762 | 0.012877 | 0.013756 |

(*) Disabled = % of population with a disability, Public Assistance = % of households receiving public assistance, Unemployed = % of the population that is unemployed, Heavy Drinkers = % of heavy drinking adults, Poor Mental Health = % of adults reporting 14+ days of poor mental health per month, Smokers = % of current adult smokers.

Table 3: Overall bias along six social dimensions for the finetuned open-source biomedical models, finetuned open-source general models, and closed-source general models averaged across 3 runs (lower scores are better). Bolded numbers depict the least biased model per dimension.

Table 2 presents the performance metrics across various models, including open-source fine-tuned biomedical models, open-source fine-tuned general models, and closed-source general models.

In the open-source fine-tuned biomedical models category, BioMistral-7B performed the best, achieving the highest scores in all metrics. M42-health and Meditron7b follow, though with slightly lower scores. The zero-shot versions of these models generally exhibit lower performance, as ex-

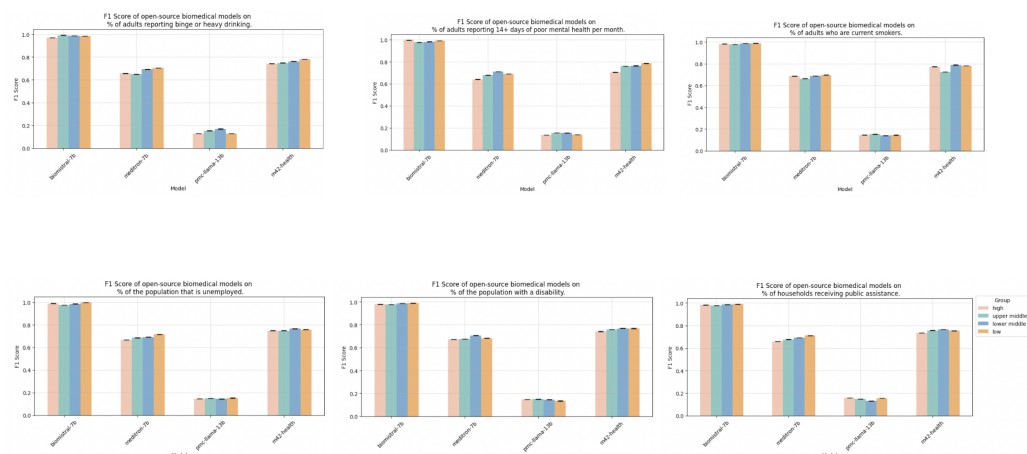

Figure 3: Disparities in performance between different groups depicted by F1 and standard deviations over 3 runs of the social groups along each dimension by each examined open-sourced biomedical LLMs.

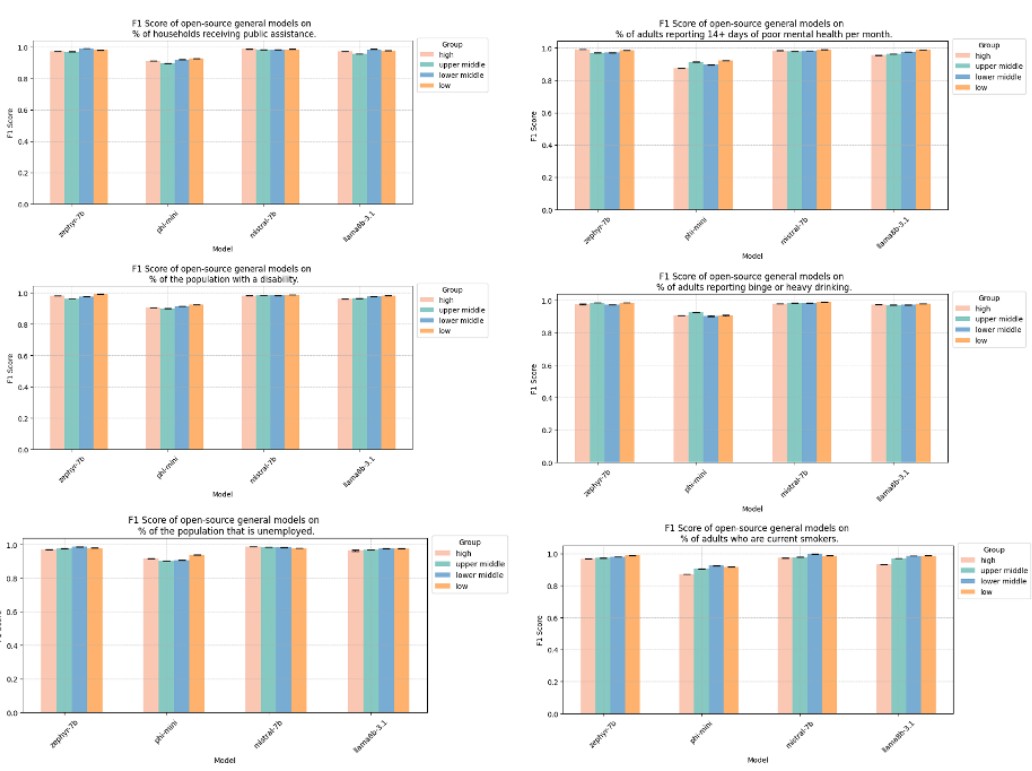

Figure 4: Disparities in performance between different groups depicted by F1 and standard deviations over 3 runs of the social groups along each dimension by each examined open-sourced general LLMs.

pected, due to the absence of task-specific fine-tuning. In the open-source fine-tuned general models, Mistral-7B performed the best, with the highest scores across all metrics. Llama-8B-3.1 and Zephyr-7B also show competitive results across all metrics. Similar to the biomedical models, the zero-shot versions of these general models generally perform lower than their fine-tuned counterparts but still maintain competitive scores in some metrics. In the closed-source general models category, Amazon-titan achieved the highest scores across all metrics. Claude, Cohere, and GPT-4o

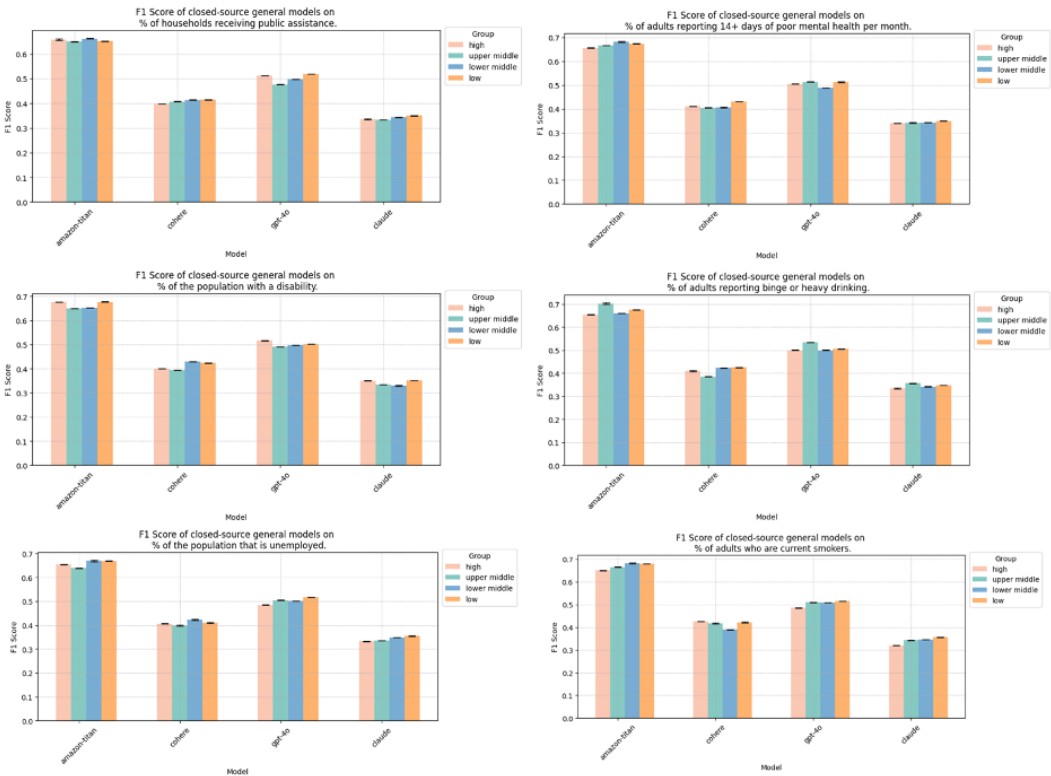

Figure 5: Disparities in performance between different groups depicted by F1 and std over 3 runs of the social groups along each dimension by each examined closed-sourced general LLMs.

demonstrated slightly lower performance. This indicates that closed-source models are competitive but often trail behind the fine-tuned open-source models in several key metrics.

Compared to baseline models, the fine-tuned models consistently perform the best across the board, highlighting the effectiveness of the fine-tuning process and validating the underlying data. The significant improvement in performance by fine-tuned models, such as BioMistral-7B and Mistral-7B, compared to zero-shot and closed-source models, underscores the value of task-specific training in improving model accuracy and generalization.

At a fine-grained level, Figure 3 plots the F1-scores for social groups across several dimensions evaluated by the biomedical models. BioMistral-7B consistently demonstrates strong performance across all groups and dimensions, with minimal disparities, in areas such as binge drinking, smoking, and mental health. Meditron-7B, while generally performing well, shows more noticeable dips in F1-scores for lower middle and low socio-economic groups, especially in dimensions like mental health and unemployment. PMC-llama-13B exhibits the most disparities, significantly underperforming for lower social groups, particularly in complex socio-economic dimensions such as unemployment, disability, and public assistance. M42-health performs similarly to BioMistral-7B, maintaining relatively equitable performance across most groups.

In Figure 4, the performance of open-source general models are evaluated based on F1-scores across different social groups. Zephyr-7B consistently shows better performance across all dimensions, maintaining minimal differences between social groups. Phi-mini similarly performs well, but there are slightly more visible differences in certain dimensions like mental health and public assistance, where lower groups show marginally lower F1-scores. Mistral-7B exhibits overall stable performance with small variations. Llama-8B-3.1 maintains high F1-scores across the board, with little to no visible performance gap between social groups.

In Figure 5 , the closed-source general models, including Amazon, GPT-4o, Cohere, and Claude, display substantial disparities in performance across social groups. These models consistently perform

better for the "high" group, while the "lower middle" and especially the "low" group experience significant drops in F1-scores. For dimensions like binge drinking and smoking, while Amazon demonstrates stronger performance, there remains a noticeable decline in the "low" group compared to higher socio-economic groups. In more complex dimensions such as mental health and disability, the disparities become more pronounced, with the "low" group falling far behind the "high" group in terms of F1-scores. Unemployment and public assistance, two socio-economic factors, show the most significant performance gaps, where closed-source models fail to maintain equitable results across all groups, especially for those in lower socio-economic categories.

Table 3 presents the performance of various open-source and closed-source models, highlighting their biases across social dimensions such as disability, public assistance, unemployment, heavy drinking, mental distress, and smoking. The models are categorized into three groups: open-source finetuned biomedical models, open-source finetuned general models, and closed-source general models.

In the open-source finetuned biomedical models category, BioMistral-7B and PMCLlama-13b outperform others with the lowest bias across most dimensions. BioMistral-7B excels in disability, public assistance , and mental distress, while PMCLlama-13b shows the least bias in unemployment and smoking. M42-health and Meditron-7B exhibit higher bias across most social dimensions. In the open-source finetuned general models, Mistral-7B shows the least bias in disability, public assistance, and unemployment with Llama-8B-3.1 having the lowest bias in heavy drinking. Zephyr-7B has competitive performance in smoking, while Phi-mini exhibits slightly higher bias overall. For the closed-source general models, Claude stands out with low bias in multiple dimensions, particularly in mental distress, public assistance, and unemployment. Amazon-titan performs well in public assistance but shows higher bias in other dimensions. GPT-4o demonstrates competitive performance, especially in disability, but exhibits higher bias in other areas compared to the top-performing open-source models.

Overall, the results suggest that open-source biomedical models generally perform better in minimizing bias in social dimensions, with BioMistral-7B and Mistral-7B leading across multiple categories. Among closed-source models, Claude and GPT-4o show the least bias, particularly in public assistance and mental distress.

## 6 CONCLUSION

In this paper, we introduce a novel and comprehensive question-answeing dataset using the MIMIC echocardiogram reports. This dataset is designed to enhance QA systems within the cardiology domain. To demonstrate the dataset's utility, we validated it using 12 LLMs, showing that the instruction fine-tuned Mistral-7B open-source model performs better than biomedical-specific models and general state-of-the-art closed-source model. Given Mistral-7B performed the best we termed it Echo-Mistral (i.e. our best fine-tuned model). Our fairness audit reveals variability in model performance across different social and marginalized communities. We hope our comprehensive benchmark, featuring multiple LLMs and various evaluation metrics, will serve as a baseline, facilitating progress in medical real-world question-answering tasks in the cardiology space.

**Limitations**. While EchoQA represents an advancement, expanding the dataset to cover a broader range of medical scenarios, both within cardiology and across the same as well as other specialties, would enhance its robustness. This expansion would also extend the framework's relevance to a wider array of medical decision-making contexts. However publicly available reports linked with social factors data is very scarce, hence we use only a single data. Additionally, in this setup, we aim to standardize the prompts to detect potential biases without skewing the system. However, it would be valuable to examine how the QA system reacts to biased prompts. Understanding its responses to biased decision-making inputs could shed light on its ability to withstand discrimination. Moreover, LLM hallucination was not investigated in this work and is left for future work.

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
