# OpenReview forum: "EchoQA: Tuning into the Heart of Echocardiogram Reports"
_ICLR.cc/2025/Conference — ICLR 2025 Conference Withdrawn Submission_

### Official Review · Reviewer_KaBc · 2024-10-27

**Soundness:** 2
**Presentation:** 3
**Contribution:** 2
**Rating:** 3
**Confidence:** 4

**Summary:**

This paper introduces EchoQA, a large-scale question-answering dataset derived from echocardiogram reports in the MIMIC database. The dataset contains 765,605 QA pairs focusing on various cardiac abnormalities. The authors evaluate multiple language models, including open-source biomedical models, general models, and closed-source models, through both fine-tuning and zero-shot approaches. They also conduct fairness audits across different social determinants of health (SDOH) to assess potential biases.

**Strengths:**

1. The work contributes a valuable and sharable benchmark and fine-tuned model for echocardiogram study.
2. Clear methodology and implementation, which is reproducible.
3. The paper includes fairness analysis which is a good implications for healthcare domain.

**Weaknesses:**

1. The novelty and significance of this work are limited. While it introduces a QA benchmark for echocardiogram reports, the dataset is merely derived from MIMIC-IV without methodological innovation. The practical value is questionable for two key reasons:
- First, converting echo reports into QA format offers little clinical utility since experienced cardiologists can already efficiently interpret these inherently structured reports.
- Second, the QA pairs primarily focus on disease abnormality detection, which essentially reduces to a diagnostic classification problem - it's unclear why the authors chose to approach this through complex LLM training rather than conventional classification models. The paper would benefit significantly from better justification of this fundamental design choice and clearer articulation of its clinical motivation.
2. A detailed breakdown of model performance across different cardiac abnormalities is encouraged to include (eg. the categories showed in Table1). This would show provide sufficient insight into how the models handle specific cardiac conditions.
3. The quality of Figure 4 need to improve.
4. The paper lacks crucial qualitative analysis, particularly regarding model failure cases. Including examples of where and why models fail would provide valuable insights for real-world deployment. Additionally, human evaluation from clinical experts would strengthen the validation of the model's performance.

**Questions:**

1. What is the reason for only sampled 10,000 QA pairs from the curated dataset?
2. How did you extract the abnormalities from the report? Could you provide more detail for this?
3. What is the potential biases in the source data?
4. Have you explored prompt engineering variations?
5. Could you specify how you calculate the F1 in Table 2?

---

### Official Review · Reviewer_edXN · 2024-10-28

**Soundness:** 1
**Presentation:** 1
**Contribution:** 2
**Rating:** 1
**Confidence:** 5

**Summary:**

The paper introduces echocardiogram reports derived 765k QAs dataset. It has tested 12 LLMs including general and biomedical models. It used social determinants of health, such as the economic status, to perform fairness audits. The result shows that the trained Mistral model perform better than the models that have evaluated. Fairness test showed open-source biomedical models are better than closed-source general models in minimizing the social bias.

**Strengths:**

1. It introduced Echocardiogram reports based QAs with the consulting of clinicians ensuring the aspect of the questions is clinically relevant.
2. It used social determinants of health to perform fairness audits and showed that the open-source biomedical models are more robust to social bias than closed-source general models.

**Weaknesses:**

1. The study uses only 10,000 QAs out of a dataset of 765,000, with 70% allocated for training. This leaves a very limited number of questions for validation and testing, which may restrict the robustness and generalizability of the findings. Utilizing a larger portion of the dataset, or adjusting the allocation to ensure adequate data for validation and testing, could provide a stronger basis for evaluating the model's performance and enhance the study's key contributions.
2. The diversity of the questions comprising the instruction tuning dataset is quite limited.  Although it is described the questions ask about various aspect, the format of the question is not diverse. As shown in Figure 2, the most of the questions are formatted as the simple questions asking is this present or absent / quantifiable. For example, some questions could have been put together as in a single prompt in order to make the prompt more complicated. Also, some efforts could have been done to make the model reason why or how it answered the presence or absence.
3. There are no samples of the generated responses shown in the paper to show how their further trained model perform better in qualitative manner. A few representative examples of generated responses from different models can be included to highlight how the fine-tuned model performs better qualitatively.
4. Although the SDOH analysis is the key contribution of the work, the paper describes this method in 4.2 section together with evaluation with lexical similarity metrics. To improve understanding, readers would benefit from more information on each discriminant and the four groups for each one. For example, a table could be provided for each social determinant, outlining the characteristics of each group. This might include details such as levels of alcohol consumption for the heavy drinker social determinant and specific diseases or conditions associated with each group for the mental health determinant.
5. The style and presentation of the work needs improvement. Table caption has to follow the ICLR2025 style (below the table not above the table). Table 3 (SDOH) has to be reformatted so that the column names are better placed. Readability of the Figures (3, 4, 5) are very low.

**Questions:**

1. Why did you do random sampling of 10,000 QAs rather than using the whole dataset?
2. Why did you group the social health attributes into 4 groups based on the quantile of the distribution rather than using a raw value? This discretization can also introduce a bias.

**Details Of Ethics Concerns:**

There is no Ethical Statement to summarize how they maintained a strict compliance with privacy regulations for the MIMIC-IV data. Although it is stated in the experimental setup section, it is still no harm to have this separate section at the end.

---

### Official Review · Reviewer_5tE6 · 2024-11-01

**Soundness:** 2
**Presentation:** 1
**Contribution:** 3
**Rating:** 5
**Confidence:** 3

**Summary:**

This paper introduces a dataset EchoQA consisting of clinician-curated question-answer pairs for a large quantity of ECG reports. This dataset could act as a useful benchmark for evaluating and training LLM systems developed as clinical aids, particularly in the cardiology domain. The authors benchmark a variety of open-source and closed-source LLMs on their dataset demonstrating large variability in performance, and they also apply fine-tuning to the open-source models leading to performance gains for most models. Finally, it was encouraging to see the authors address issues of algorithmic bias, comparing performance across different demographic sub-groups in different socioeconomic domains, demonstrating inequitable performance for several LLMs.

**Strengths:**

EchoQA may be an important contribution to the small but growing set of benchmark datasets in the medical domain, which will be essential if LLMs are going to be adopted in the clinic. Additionally, the focus on algorithmic bias helps highlight important issues associated with the usage of LLMs in clinical applications and helps to normalise fine-grained LLM evaluations that go beyond simple pointwise metrics.

**Weaknesses:**

This manuscript lacked important details about how EchoQA was constructed making it difficult to evaluate the validity and usefulness of this dataset.

Please see the questions for areas where more clarity is needed.

**Questions:**

For dataset creation:
1. How did clinicians generate questions? Were a single set of questions generated that were applied across all reports or were questions generated on a per-report basis. If the former, then how many questions were there in total, or were there the same number of questions as answers.
2. How many clinicians did you use to generate questions?
3. What was the process used to generate answers? Please explain this text analysis pipeline in full detail. Did it use an LLM or were the answers generated through text parsing?
4. How can the validity of the automated answers be verified? Did the authors or clinicians involved perform any spot-checks on a sample to ensure the answers were in fact correct? Were there any other measures introduced to ensure data quality?
5. It would also be useful to have some more general statistics on the QA pairs. How long are the answers (in words or tokens)?

For model evaluation:
1. Why were only 10,000 pairs used for model evaluation?
2. The evaluation results are all based on ROUGE/BLEU/METEOR/F1. These metrics may not necessarily capture richer semantic information, which could lead to inaccurate results, especially for more complicated, longer answers. Did the authors conduct any additional analyses to verify that these metrics did indeed accurately identify correct answers?
3. The fairness audit analysis approach was unclear. Are the results reported for questions in the dataset related to the social dimensions? How were these social dimensions and social groups extracted from the text?

Additional minor comments:
* In the introduction the authors mention that the large volume of and complexity of ECG reports represents a challenge for clinicians. Do the authors have any references or supporting information to back up this claim?
* Figure 1 is not very informative, and it is hard to understand how the dataset was constructed based on this figure alone. In line with the additional information requested for understanding dataset construction, more detail in this figure would also be helpful.
* Figure 2 seems to suggest that QA pairs were both annotated by clinicians, but is it not the case that the question was generated automatically? If so the figure should be updated to make that clear.
* For figures 3-5, though I think the results across the different social groups are interesting, I think the key contribution from this paper is the dataset. As such these results would be better suited to the appendix with more space from the main body dedicated to explaining how the dataset was constructed.
* There is a typo “GPTo” on line 211? Assume they mean GPT4o?

---

### Note · Authors · 2024-11-13

I have read and agree with the venue's withdrawal policy on behalf of myself and my co-authors.